# Multitask feature learning approach for knowledge graph enhanced recommendations with RippleNet

YueQun Wang[1], LiYan Dong[1,2], YongLi Li[3‡], Hao Zhang[1,2‡]*

1 College of Computer Science and Technology, Jilin University, Changchun, China, 2 Key Laboratory of Symbolic Computation and Knowledge Engineering of Ministry of Education, Jilin University, Changchun, China, 3 School of Computer Science and Technology, Northeast Normal University, Changchun, China

☯ These authors contributed equally to this work.
‡ These authors also contributed equally to this work.
* zhangh@jlu.edu.cn

**Data Availability Statement:** Data is available at https://github.com/hwwang55/MKR/tree/master/data.

**Funding:** The authors received no specific funding for this work.

## Abstract

Introducing a knowledge graph into a recommender system as auxiliary information can effectively solve the sparse and cold start problems existing in traditional recommender systems. In recent years, many researchers have performed related work. A recommender system with knowledge graph embedding learning characteristics can be combined with a recommender system of the following three forms: one-by-one learning, joint learning, and alternating learning. For current knowledge graph embedding, a deep learning framework only has one embedding mode, which fails to excavate the potential information from the knowledge graph thoroughly. To solve this problem, this paper proposes the Ripp-MKR model, a multitask feature learning approach for knowledge graph enhanced recommendations with RippleNet, which combines joint learning and alternating learning of knowledge graphs and recommender systems. Ripp-MKR is a deep end-to-end framework that utilizes a knowledge graph embedding task to assist recommendation tasks. Similar to the MKR model, in the Ripp-MKR model, two tasks are associated with cross and compress units, which automatically share latent features and learn the high-order interactions among items in recommender systems and entities in the knowledge graph. Additionally, the model borrows ideas from RippleNet and combines the knowledge graph with the historical interaction record of a user's historically clicked items to represent the user's characteristics. Through extensive experiments on real-world datasets, we demonstrate that Ripp-MKR achieves substantial gains over state-of-the-art baselines in movie, book, and music recommendations.

## 1 Introduction

Recommender systems are known as the growth engine of the Internet. A better recommender system model can facilitate users in efficiently obtaining high-interest information under the circumstance of information overload, improve the user conversion rate of products, and

**Competing interests:** The authors have declared that no competing interests exist.

achieve the purpose of continuous growth of a company's business objectives. To improve the accuracy of a recommender system, a large amount of side information is added to the model of the recommender system to extract the hidden content of the information and to enhance the association between users and projects. Side information can be understood as auxiliary information, such as item attributes [1, 2], item reviews [3, 4], and users' social networks [5, 6].

Because of its characteristics, a knowledge graph is more suitable for correlation mining than causality mining. By deeply mining the deep relationships among projects, users, and between projects and users from the knowledge graph, more relevant results can be obtained, which is conducive to personalized recommendation for users, improving the diversity of recommendation results and maintaining a high recommendation accuracy.

The recommender model with knowledge graph as the edge information can mine the information of knowledge graph and combine it with the existing user information and project information to enhance the dimension of data and thus improve the accuracy of recommendation. Joint learning and alternating learning are two kinds of learning methods that combine knowledge graph and recommender system, but each has its advantages and disadvantages. For example, The RippleNet [7] model represents joint learning pays more attention to the user-item interaction matrix while neglecting the knowledge graph's structural information. In RippleNet, the knowledge graph's relational elements are weakly characterized because R's embedded matrix is challenging to be trained to capture the quadratic. MKR pays attention to the knowledge graph's structural information for alternating learning mode and uses a cross and compression unit to connect the recommendation module and knowledge graph module to carry out training in a multi-task way. However, it also ignores the critical information carried by the project scoring matrix. Therefore, this paper combines the two training methods to maximize knowledge graph information mining.

To obtain the potential information in a knowledge graph and maximize the information content of a knowledge graph, we apply the Ripp-MKR model, which combines the preference propagation idea of RippleNet with the cross-training idea of the MKR model. KG side information is combined with the history of user interaction information to represent the user's feature vector, and the KG is again used as side information with item one-hot embedding to train the maximum digging KG of the hidden information. We are using the cross-compression unit's idea in the MKR model, deeper mining of knowledge graph is carried out. The tail vectors (item attribute information) in the knowledge graph are updated with this unit. The user vector can be represented by the historical item attribute (the knowledge graph's tail vector). The item vector can be represented by the head vector of the knowledge graph. This process is iterated, and the user vectors and item vectors are updated at the same time.

The significant difference between Ripp-MKR and the existing literature is that Ripp-MKR combines joint learning and alternating learning in two ways for the process of knowledge graph feature learning applied to recommender systems: (1) The KGE methods are incorporated into recommendations by preference propagation to use the information and the preference information to represent a user's feature vector. (2) The knowledge graph and recommendation algorithm's feature learning is regarded as two separate but related tasks, and the framework of multitask learning is used for alternating learning.

In summary, our contributions in this paper are as follows:

1. To the best of our knowledge, this is the first work to combine joint learning and alternating learning methods in KG-aware recommendations.

2. We propose Ripp-MKR, a framework utilizing KGs to assist recommender systems. Ripp-MKR automatically discovers users' hierarchical potential interests by iteratively

propagating users' preferences in a KG, and the recommendation module and the KGE module are bridged by a specially designed cross and compress units.

3. We conduct experiments on three real-world recommendation scenarios, and the results prove the efficacy of Ripp-MKR over several state-of-the-art baselines.

We preserved the historical information of user interaction in the RippleNet model through the above work, and we also preserved the knowledge graph structure mining in the MKR model. The knowledge graph is fully mined in the Ripp-MKR model. Through the user's historical clicks and knowledge graph information, the user's preferred item attribute set is obtained, and the item attribute set is used to represent the user, According to the overlap between items and entities in the knowledge graph, cross and compression unit is used for multi-task training to ensure the maximization of knowledge graph structure mining. In our proposed Ripp-MKR, under the condition that the amount of inherent information remains unchanged, the prediction accuracy and recall rate can be maximized by adding less time and space complexity.

## 2 Related work

Since the user set, item set, and user score matrix of a recommender system are highly integrated and correlated with a knowledge graph, many scholars are keen to study knowledge graphs as recommendation models for side information.

When side information is used to learn about item embedding, an item of the same brand or category should be more similar. Recommendation algorithms can be divided into two categories according to data types: the first category is recommendation algorithms based on user behavior data, also known as collaborative filtering. Collaborative filtering can be divided into two categories: memory-based and model-based. A representative algorithm of memory-based collaborative filtering includes UserCF, based on a user, and ItemCF, based on an item, and their function is to directly calculate the similarities between user-user or item-item and behavioral data. The representative algorithms of model-based collaborative filtering mainly include some implicit variable models, such as SVD, matrix factorization (MF) [8, 9], the PLSA theme model, and LDA [10]. These models use behavioral data to calculate the implicit vectors of users and items and then calculate the matching degree between user-user or item-item to make recommendations. In the second class of algorithms, the most common model is the CTR model. The CTR model is essentially a binary classifier that commonly uses LR, XgBoost [11], lightGBM [12], and other classifiers.

For the two types of models, different side information is used to improve the recommendation's accuracy. For the first type of algorithm, in addition to the behavioral data of users, portrait data of users and objects can also be used, such as gender, age, region, label, classification, title, and text. For the second type of algorithm, behavioral data and side information are used to construct training samples' characteristics and class criteria.

In addition to the attribute data mentioned above, other data structure information can also be used as side information, including social networks [13], attributes [14], multimedia (e.g., texts [15] and images [16]), and knowledge graphs (KGs).

Proposed in this paper, the recommender system based on knowledge graph as side information, the structure of knowledge graph and content can be better integrated into the recommender system in user-item interaction matrix. On the one hand, to build a knowledge graph and use its internal structure and item attribute node to represents the item vectors, on the one hand, the user vector is represented by the user's history score combined with the knowledge graph. Combines knowledge graph and recommender system because the topology of the

knowledge graph and recommender system in the item attributes, user, and user rating matrix can be the perfect fusion. Knowledge graphs, as side information based on traditional data mining, are more likely to dig up its internal topology information.

## 2.1 Side information

To solve the sparse data and cold start problems of a recommender system, many recommendation algorithms have been proposed, which utilize the profile information of users or items (such as social network and item category) and have high effectiveness in improving the recommendation performance. In recent years, a recommender system not only uses the interactive historical data of users and items but also adds considerable side information to estimate users' preferences.

Side information can be divided into two categories: structured data and nonstructured data. Structured data can be divided into flat features, feature hierarchies, and knowledge graphs. For nonstructured data, text features, image features, and video features are included. The above data can be integrated into a recommender system through a unique deep learning training model, which has also attracted the researchers' attention. He and Xiangnan et al. [17] modeled the implicit feedback information (interaction matrix) of user projects and improved the traditional collaborative filtering model. The multilayer perceptron (MLP) is used to model the user-project interaction matrix. Man, Tong et al. [18] proposed a cross-domain recommended embedded mapping framework called EMCDR. The proposed EMCDR framework is different from the existing cross-domain recommendation model in two aspects. First, embedding learning is realized by employing the implicit factor model in each domain to learn the entities' specific features in each domain. Second, a graphing technique is used to supplement sparse data in different domains. Zheng et al. [19] proposed using two parallel CNNs to process text information corresponding to comments at the user level and comments at the project level to mine users' behavioral characteristics and the attribute characteristics of projects. The ConvMF model proposed by Kim et al. [20] also uses text information to alleviate the scoring matrix's sparsity by using text information as auxiliary edge information. The Wide&Deep learning model proposed by Cheng et al. [21] uses the characteristics of users and projects to improve recommendation accuracy. For the Ripp-MKR model presented in this paper, KG is mainly used to fuse the recommender system's side information.

## 2.2 KGE for RS

A recommender system refers to analyzing historical data to recommend products that users are likely to like or buy, the core of which is the users and items. Furthermore, the key to recommender systems includes three parts: user preference, item features, and interactions.

The common approaches for embedding-based methods are divided into two steps. The first step is to embed all entities and relationships with knowledge graph embedding (KGE) technology, such as TransE [22], TransR [23], TransH [24], and other Trans series algorithms. The second step is to fuse the entity and relationship embedded vectors with item-user interactions and then model the knowledge graph.

For instance, Zhang et al. proposed CKE [25], which unifies various side information types in the CF framework. They fed an item's structural knowledge (items attributes represented with a knowledge graph) and content (textual and visual) knowledge into a knowledge base embedding module. First, they adopted TransR to calculate the embedding knowledge graph, and each entity embedding was extracted as the structural vector of the item. Then, the SDAE model was adopted to obtain the descriptive text of the items. Finally, the SCAE model was adopted to obtain the visual embedding of the item-related images. Therefore, an item

representation vector is the sum of the original vector and the new information vector from the three aspects. However, CKE does not utilize the relationship information among entities in a knowledge map, and user embedding has not been updated. Even though the additional information is multifaceted, adding a layer of weight could be beneficial.

Wang et al. proposed DKN [26] for news recommendations. It models the news by combining the textual embedding of sentences learned with Kim's CNN [27] and the knowledge-level embedding of news content entities via TransD. DKN introduces a knowledge graph to measure user preferences based on the user's past interactions. However, only one hop is selected as the analysis of context information, and the relational information in the map is not involved at all.

The other type of embedding-based method directly builds a user-item graph, where users, items, and their related attributes function as nodes. Zhang et al. proposed CFKG [28], which constructs a user-item KG. User behaviors (purchases and mentions) are regarded as one relation type among entities in this user-item graph. Multiple types of item side information (review, brand, category, bought together, etc.) are included. Wang et al. proposed SHINE [29], which takes the celebrity recommendation task as the sentiment link prediction task among entities in a graph. Besides, this model uses social networks, personal data information, and sentiment networks as side information. Dadoun et al. proposed DKFM for POI recommendations. DKFM [30] applies TransE over a city KG to enrich the representation of a destination.

Yang et al. introduced a GAN-based model, KTGAN [31], for movie recommendations. In the first phase, KTGAN learns the knowledge embedding for movies by incorporating the Metapath2Vec model [32] with a movie's KG and tag embedding with the Word2Vec model [33] on a movie's attributes. Later, Ye et al. proposed BEM [34], which uses two types of graphs for items, the knowledge-related graph (containing item attribute information, such as brand and category) and behavior graph (containing item interaction-related information, including cobuy, corate, and coadd to cart) for recommended. In Hongwei Wang's paper on MKR [35] (multitask feature learning for knowledge graph enhanced recommendations), the main idea is that there is overlap among items in a recommender system and entities in a knowledge graph, multitask learning frameworks, two separate but related tasks, and alternating learning. MKR's model framework includes recommender systems tasks and knowledge graph feature learning tasks. The recommended inputs are characteristic representations of users and items, and the estimated click-through rate is the output. In the feature learning part of the knowledge graph, the head node and relation of triples are used as input, and the predicted tail node is used as output. The MKR model applies a pattern of alternating training between the knowledge graph and recommender systems.

Path-based methods are primarily based on the inherent correlation structure of knowledge graphs, and the recommended pattern, namely, "metapath ", is defined artificially in advance. For example, in a movie recommendation task, a metapath can be defined as a "user-item-actor-item" or "user-item-director-item" and then modeled on the metapath.

Xiao Yu, Xiang Ren, Ycursed Sun, et al. proposed the PER [36] model published in 2014. In this paper, the concept of a metapath was proposed. Based on each type's predefined paths, the authors extracted all paths that fit the definition of the metapath for each pair of user items and designed an evaluation function to measure the possibility of user-item interaction. Then, the classic MF algorithm is used to model and solve for the user and item vectors. Similar to PER, Huan Zhao et al. [37] designed metapaths for Yelp and Amazon datasets and obtained L evaluation functions. However, the FM model is adopted in the modeling, and the FM-group lasso is used for further optimization. Sun et al. proposed a recurrent knowledge graph embedding (RKGE) approach that mines the path relation between a user and an item automatically,

without manually defining metapaths. Wang et al. proposed a knowledge-aware path recurrent network (KPRN) [38] solution. KPRN constructs the extracted path sequence with both the entity embedding and the relation embedding.

For unified methods, the RippleNet model is the leading representative and the basis for improving this article. RippleNet proposed that users' preferences spread like waves on the knowledge graph, which is also known as one-hop, two-hop, etc. For an embedding knowledge graph, user embedding is realized through a hop communication of an item that has interfaced with the user. In other words, starting from an item entity with historical interactions, embedding is diffused into three layers, and an entity embedding weighted sum is realized at each layer. Finally, an embedding of the three layers is added.

## 3 Formulation

We formulate the knowledge graph enhanced recommendation problem in this paper as follows. The recommender systems mainly include user set $U$ and item set $V$. The user set can be represented as $U = \{u_1, u_2. \ldots u_n\}$. Similarly, the sets of items can be expressed as $V = \{v_1, v_2. \ldots v_m\}$. The user-item interaction matrix composed of n users and m items can be expressed as $Y \in R^{m \times n}$, where $y_{uv} \in Y = 1$ indicates user $u$ engaged with item $v$; otherwise, $y_{uv} \in Y = 0$. The knowledge graph $G$ is generally represented by triples $(h,r,t)$. Here, $h \in E$, $r \in R$, and $t \in E$ denote the head, relation, and tail of a knowledge triple. $E$ and $R$ denote the set of entities and relations in the KG. For example, the triple (*Jack Chan*, *actor*, *Rush hour*) means Jackie Chan and Rush Hour are two entities, and *film.actor.film* is the relationship between the two entities. Thus, it is easy to find multiple KG triples between entities associated with the knowledge graph during the recommendation process. The relationship structure between the knowledge graph and the recommender system is shown in Fig 1. The notations we will use throughout the article are summarized in Table 1.

Given user-item interaction matrix Y as well as knowledge graph G, we aim to predict whether user u has a potential interest in item v with which he has had no interaction before. Our goal is to learn a prediction function $\hat{y}_{uv} = F(u, v; \Theta)$ where $\hat{y}_{uv}$ denotes the probability that user u will click item v, and $\Theta$ represents the model parameters of function F.

## 4 RIPP-MKR model

In this section, the main structure of the Ripp-MKR model is introduced. The mathematical representation of related terms in the knowledge graph and recommender system presents the representation of user feature vectors and the interaction between multitask units.

### 4.1 Framework

The framework of the Ripp-MKR model is shown in Fig 2. The framework can be divided into a recommendation module, KGE module, and cross and compress units framework.

As shown in Fig 2, the tail vectors are predicted by the knowledge graph's head vectors and the relation vectors (represented by the blue module). The tail vectors represent the attribute values of the item. According to users' historical interaction information, perform ripple propagation gets user preferences for item's attributes, aggregate these tail vectors, and form a preliminary user vector representation (green module representation). Through ripple propagation in different hops, multiple user vector representations can be obtained and merged to form the final user vector (the rightmost part of the figure). During this period, the cross-training of item vectors (the leftmost part of the figure) and head vectors in the knowledge graph is performed through the cross-compression unit, and the knowledge graph and the recommender system are merged again to iterate the above process. On the one hand, the

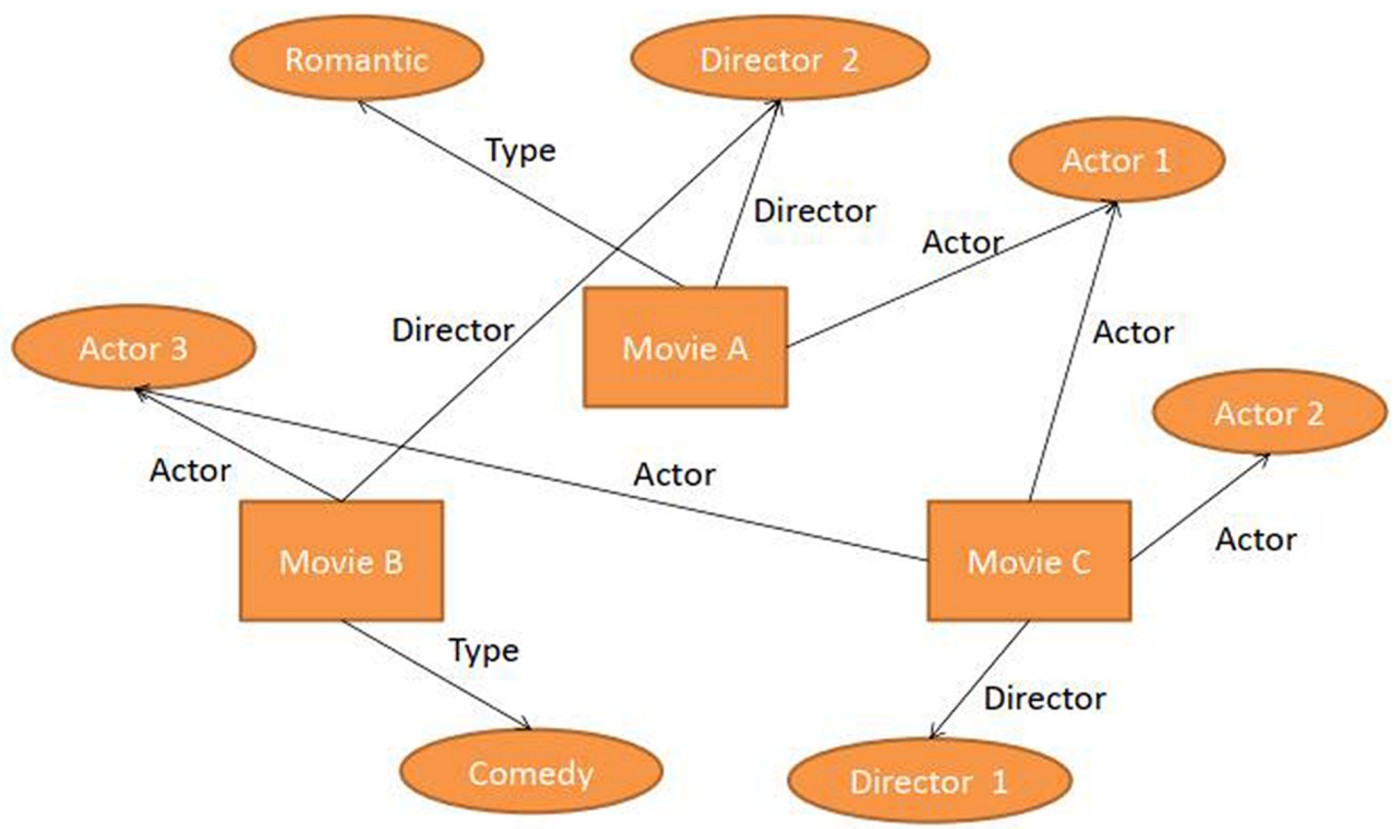

**Fig 1. The relationship structure between the knowledge graph and recommender system.**

user vector is represented by the knowledge graph and the user's historical interaction information. Then cross-training is performed by combining the head vector of the knowledge graph with the item vector in the recommender system. Converged by the loss function, and the final vector representation is obtained and recommended.

The KGE module uses multiple layers to extract features from the head and relation of a knowledge triple. The predicted tail characteristics can be output under the control of the loss function. In this paper's model, the training of the KGE module is similar to that of the MKR model, while for the recommended module, the training is improved based on the MKR model. In the MKR model, the feature learning of users and items is extracted through multiple layers. In the Ripp-MKR model, the idea of RippleNet is utilized to obtain users' interest propagation lists through the historical interaction information between users and items. In this process, the knowledge graph's t vector is combined to express users' feature vectors instead of the one-hot method. The cross and compress units are another bridge between the KGE module and the recommendation module; they can automatically learn the high-order feature interactions of items in recommender systems and entities in the KG.

## 4.2 Cross and compress unit

The method of the MKR model combining the KG model and RS model has been described previously. For the Ripp-MKR model, the cross and compress units are represented by a red rectangle in Fig 2, which will be described in this section. The cross and compress unit generates a cross feature matrix from item and entity vectors by cross operation, and outputs their vectors for the next layer by compress operation.

**Table 1. Notations and explanations.**

| Notations | Descriptions |
|---|---|
| $U$ | user set |
| $V$ | item set |
| $u_i$ | user i |
| $v_j$ | item j |
| $\boldsymbol{u}_i \in R^d$ | raw feature of user i |
| $\boldsymbol{v}_j \in R^d$ | raw feature of item j |
| $\boldsymbol{u}_l$ | latent feature of user u |
| $\boldsymbol{v}_l$ | latent feature of item v |
| $\mathbf{V_u}$ | User u's interaction history set |
| $Y \in R^{m \times n}$ | user-item interaction matrix composed of n users and m items |
| $y_{uv} \in Y$ | implicit feedback from user u to item v |
| $\hat{y}_{uv}$ | Prediction function the probability that user u will click item v |
| $G$ | Knowledge graph |
| $E$ | Entity set of knowledge graph |
| $R$ | Relation set of knowledge graph |
| $h \in \boldsymbol{E}$ | Head vector |
| $r \in \boldsymbol{R}$ | Relation vector |
| $t \in \boldsymbol{E}$ | Tail vector |
| $\hat{t}$ | The predicted vector of tail t |
| $\boldsymbol{C}_l \in R^{d \times d}$ | Cross feature matrix of layer L |
| $\boldsymbol{C}^L$ | The cross and compress unit which have L layers |
| $\boldsymbol{e}_l \in R^d$ | latent feature of entity $e$ |
| $[\boldsymbol{e}]$ | The entity e obtained by Cross and Compress Unit |
| $[\boldsymbol{v}]$ | The item v obtained by Cross and Compress Unit |
| $\boldsymbol{v}_l^{(i)} \boldsymbol{e}_l^{(j)}$ | Possible feature interaction between item v and its associated entity e is modeled explicitly in the cross feature matrix of layer $L$. |
| $\boldsymbol{w}_i \in R^d$ | trainable weight |
| $\boldsymbol{b}_i \in R^d$ | bias vectors |
| $S(h)$ | The associated items of entity h |
| $S(v)$ | The associated items of item v |
| $\boldsymbol{M}^K$ | A fully connected neural network with K layer |
| $f_{KG}()$ | Score (similarity) function for KG |
| $\mathcal{E}_u^k$ | The set of k-hop relevant entities for user u |
| $S_u^k$ | The set of knowledge triples that are k-hop(s) away from seed set $V_u$. |
| $P_i$ | relevance probabilities |

The cross and compress units are shown in Fig 3. This unit is the link module between item **v** and one of its associated entities **e**. For the latent feature of latent feature $\boldsymbol{v}_l \in R^d$ and the latent feature of latent feature $\boldsymbol{e}_l \in R^d$, we construct $\boldsymbol{C}_l$, representing the cross feature matrix of layer $L$. $\boldsymbol{v}_l^{(i)} \boldsymbol{e}_l^{(j)}$ is the possible feature interaction between item v and its associated entity e is modeled explicitly in the cross feature matrix of layer $L$.

$$\boldsymbol{C}_l = \boldsymbol{v}_l \boldsymbol{e}_l^T = \begin{bmatrix} \boldsymbol{v}_l^{(1)} \boldsymbol{e}_l^{(1)} & \cdots & \boldsymbol{v}_l^{(1)} \boldsymbol{e}_l^{(d)} \\ \vdots & & \vdots \\ \boldsymbol{v}_l^{(d)} \boldsymbol{e}_l^{(1)} & \cdots & \boldsymbol{v}_l^{(d)} \boldsymbol{e}_l^{(d)} \end{bmatrix} \tag{1}$$

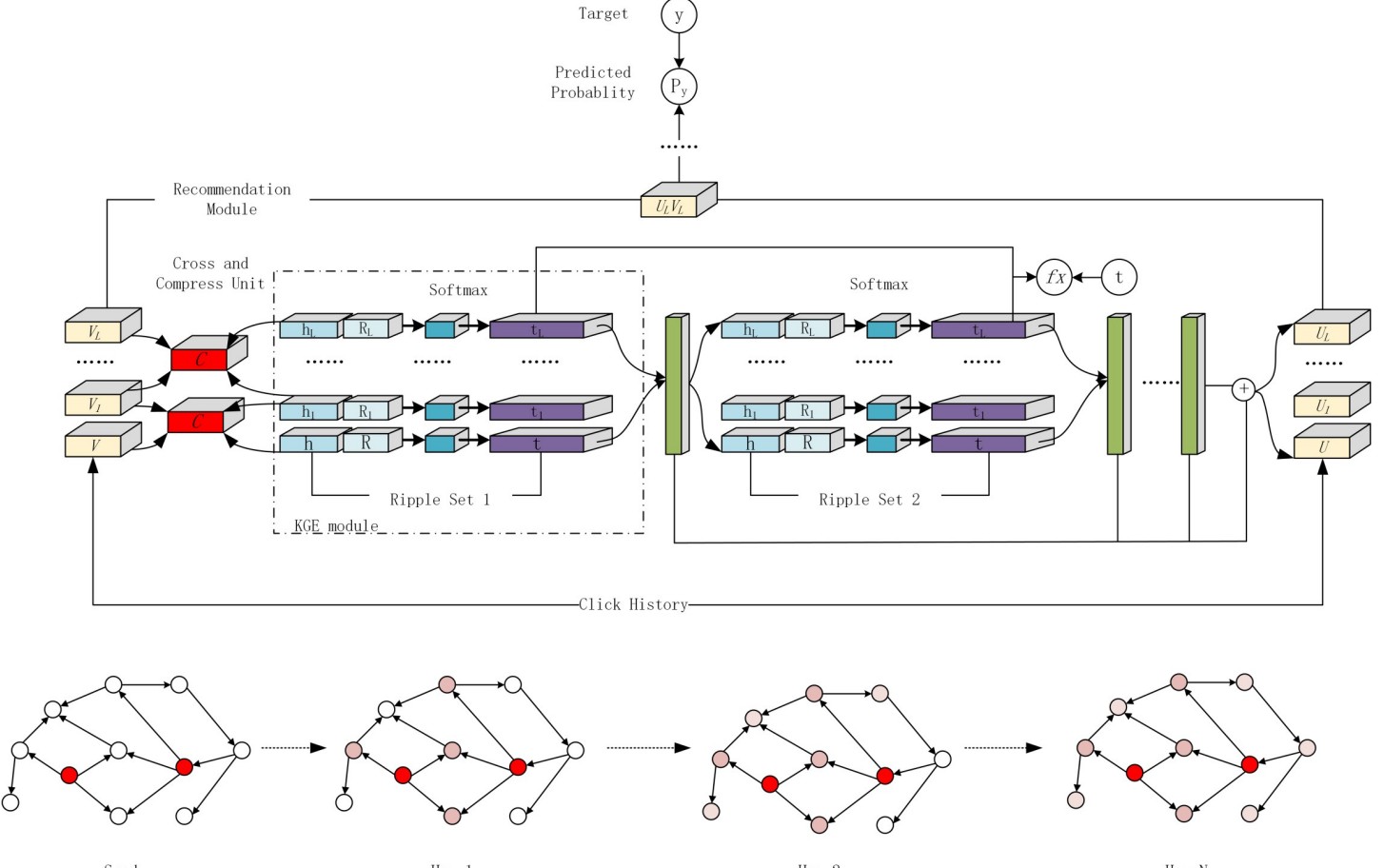

**Fig 2. The framework of Ripp-MKR.**

Eq 1 describes the cross operation in cross and compress units. In cross and compress units, there are compress operations in addition to cross operations. The feature vectors of items and entities for the next layer are output by projecting the cross feature matrix into the latent representation spaces, as shown in Eq 2.

$$\mathbf{v}_{l+1} = \mathbf{C}_l w_l^{VV} + \mathbf{C}_l^T w_l^{EV} + b_l^V$$

$$e_{l+1} = \mathbf{C}_l w_l^{EV} + \mathbf{C}_l^T w_l^{EE} + b_l^E \tag{2}$$

where $w_i \in \mathrm{R}^d$ and $b_i \in \mathrm{R}^d$ are trainable weight and bias vectors. The cross and compress unit can be denoted as the formula:

$$[\mathbf{v}_{l+1}, \mathbf{e}_{l+1}] = \mathrm{C}(\mathbf{v}_l, \mathbf{e}_l) \tag{3}$$

We use a suffix [**v**] or [**e**] to distinguish its two outputs in this paper's following. Through cross and compress units, Ripp-MKR can adaptively adjust the weights of knowledge transfer and learn the relevance between the two tasks.

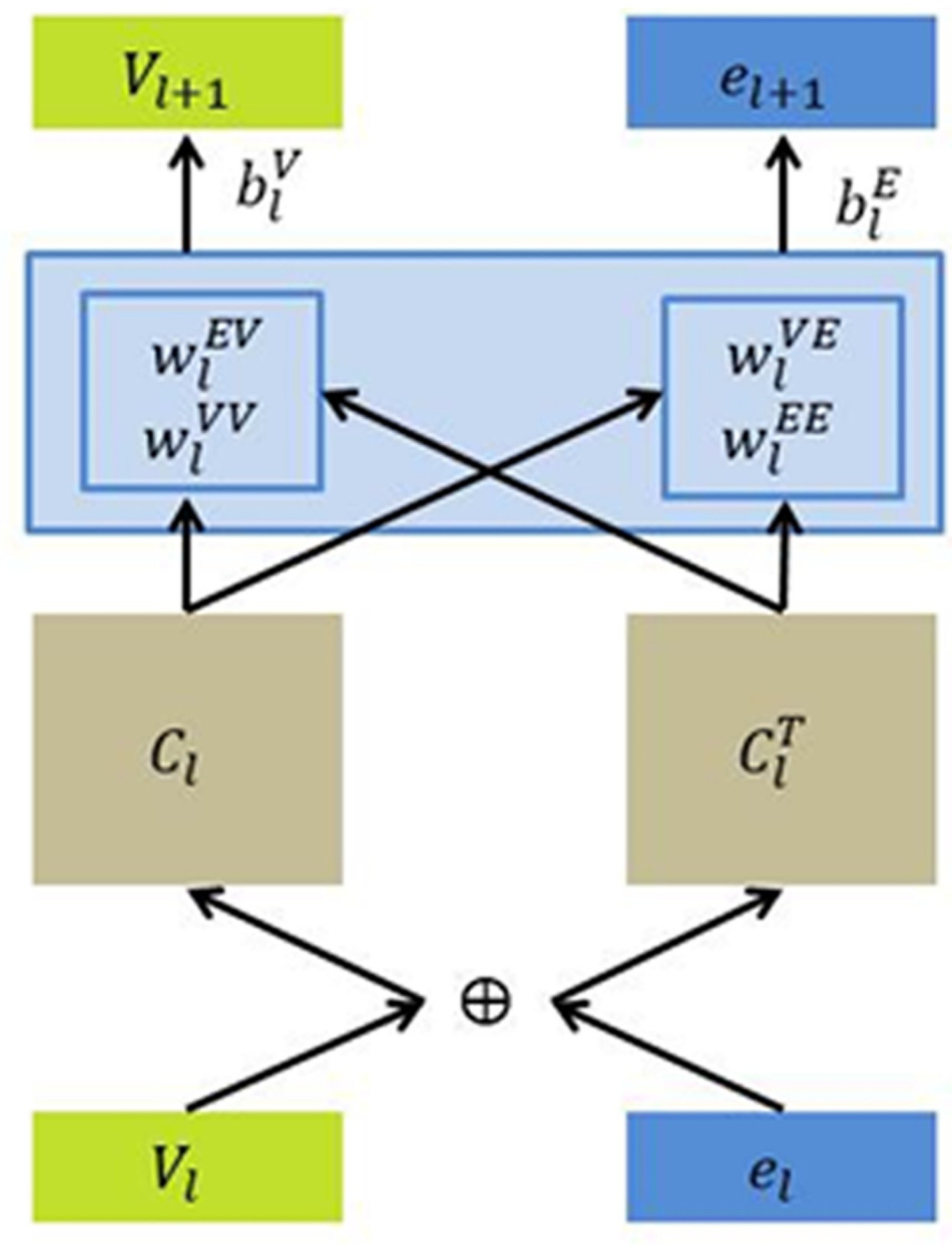

**Fig 3. The framework of cross and compress units.**

### 4.3 KGE modul

The purpose of knowledge graph embedding is to represent the entities and relationships as vectors and map the KG triples to low-dimensional space under the supervision of functions while preserving their structure. In the recommendation model design process, a knowledge graph is often used as side information because a triple of the knowledge graph has a high correlation with the items and users of the recommender system. When using a KG as side information for the recommender system, item attribute characteristics can be represented as triples of the knowledge graph.

In the KGE module of Ripp-MKR, a head vector and a relation vector are taken as inputs. The head and relation features are extracted using the multilayer perceptron (MLP) and cross and compress units, respectively. The head embedding of the KG triple corresponds to the item ID in the recommender system. The relation embedding corresponds to the item attribute, and the tail embedding corresponds to the specific item attribute value. The process of obtaining a k-layer MLP for predicting tail t is as Eq 4:

$$\boldsymbol{h_L} = E_{v \sim S(h)}[\boldsymbol{C^L}(\boldsymbol{v}, \boldsymbol{h})[\boldsymbol{e}]]$$

$$\boldsymbol{r_L} = \boldsymbol{M}(\boldsymbol{M}(\boldsymbol{M}(\cdots \boldsymbol{M}(\boldsymbol{r})))) = \boldsymbol{M^L}(\boldsymbol{r})$$

$$\hat{t} = \boldsymbol{M^K}\left(\begin{bmatrix} h_L \\ r_L \end{bmatrix}\right) \tag{4}$$

$S(h)$ stands for the association set of h in the knowledge graph, $\mathbf{v}$ stands for the item ID corresponding to $\mathbf{h}$ in the recommender system's data, and $\boldsymbol{C^L}$ is the cross and compress unit. M (x) = σ(Wx+b) is a fully connected neural network layer with weight $\mathbf{W}$, bias $\mathbf{b}$, and nonlinear activation function σ(·). For simplicity, we use the exponent L in Eq (4) and the following equation throughout the rest of this paper but note that the L layer parameters are different. $\boldsymbol{M^K}$ is a fully connected neural network with K layer. And $\hat{t}$ is the predicted vector of tail t. $\boldsymbol{h_L}$ and $\boldsymbol{r_L}$ are respectively latent features of h and r. Finally, the score of triple (h, r, t) is calculated using a score (similarity) function $f_{KG}$.

$$score(h, r, t) = f_{KG}(\mathrm{t}, \hat{\mathrm{t}}) \tag{5}$$

In this model, $f_{KG}$ is defined in the form of the inner product using the same treatment as in the MKR model:

$$f_{KG}(t, \hat{t}) = \sigma(t^{\top}, \hat{t}) \tag{6}$$

### 4.4 Recommendation modul

The Ripp-MKR model of the KGE module part is consistent with the conventional MKR model, and no great innovation has been made In contrast, the recommendation module is the main innovation and improvement of this paper. Among the models that take knowledge graphs as side information, knowledge graphs are mainly carried out in two ways: joint training or alternating training. The Ripp-MKR model integrates these two methods, which is bound to improve the mining degree of latent information of the knowledge graph.

The input of the recommendation module in MKR consists of two primary feature vectors, u and v, that describe user u and item v. In the input process, a one-hot method is used to encode the user and the item by transferring the V vector and the E vector of the KGE model

to the cross and compress unit and the recommender system's deep fusion, and the knowledge graph is completed.

The Ripp-MKR model proposed in this paper combines the knowledge graph processing method of the RippleNet model with the knowledge graph processing method of the MKR model. The RippleNet model uses t vectors combined with the user's interaction history and the items and represents the user's vectors through the t vector set of the knowledge graph. As an improvement point, the ripple propagation idea of RippleNet is applied to the user representation method of the recommendation module in the MKR model to eliminate the original one-hot coding.

The Ripp-MKR model uses user U's interaction history $\mathbf{V_u}$ and item V as input in the recommendation module, t. Start with the item of user U's history click as the seed collection (using these items as the user's existing preference information). The following logic exists:

$$\mathbf{V_u} \subseteq \mathbf{V} \tag{7}$$

Therefore, any member of $V_u$ must be a member of V. Therefore, $V_u$ is a subset of V. In a KG, each item has its corresponding triplet (item, attribute, and attribute value), so the user's item interaction history set can be changed into the set of corresponding attribute values, which correspond to the t vector in the knowledge graph.

For the input user u, the historical set of interests $V_u$ is treated as seeds in the KG and then extended along links to form a set of k-hop relevant entities for user u $\mathcal{E}_u^k (k = 1, 2, \ldots, H)$. The relational entity of user U can be defined as:

$$\mathcal{E}_u^k = \{t | (h, r, t) \in G \text{ and } h \in \mathcal{E}_u^{k-1}\},$$

$$k = 1, 2, \cdots, H \tag{8}$$

When the value of k is 0, the value of $\mathcal{E}_u^k$ is the collection of historical interaction items of user U, which can be seen as the seed set of user u in the KG.

The k-hop ripple set for user U is defined as a related triplet with k−1 relevant entities as heads. $S_u^k$ is the set of knowledge triples that are k-hop(s) away from seed set $\mathbf{V_u}$. Consistent with the RippleNet model, we can define the ripple set as:

$$\boldsymbol{S_u^k} = \{(h, r, t) | (h, r, t) \in G \text{ and } h \in \mathcal{E}_u^{k-1}\},$$

$$k = 1, 2, \cdots, H \tag{9}$$

Through this recursive form, ripple simulation is carried out; water wave transmission is carried out in the knowledge graph. The transferred entity sequence is composed of the set of knowledge graphs, and the corresponding user vector representation is completed. To avoid the ripple set being too large, a maximum length is usually set as a cutoff. On the other hand, the constructed knowledge graph is a directed graph that considers only the degree of the point's output.

As shown in Fig 2, user embedding is realized by adding vectors represented by the green rectangle in the figure. The vector represented by the first green rectangle needs to use the ripple set of 1-hop (the first item set that extends outward in the KG). Given the item embedding $\boldsymbol{v}$ and the 1-hop ripple set $S_u^1$ of user u, each triple($h_i$, $r_i$, $t_i$) in $\boldsymbol{S_u^1}$ is assigned a relevance probability $p_i$ by comparing item $\boldsymbol{v}$ to the $\mathbf{h_i}$ and $\mathbf{r_i}$ in this triple. The relevance probabilities are as

follows:

$$p_i = \text{softmax}(v^T R_i h_i)$$

$$= \frac{\exp(v^T R_i h_i)}{\sum_{(h,r,t) \in S_u^1} \exp(v^T R_i h_i)} \tag{10}$$

Finally, the embedding vector represented by the first green rectangle is weighted for all corresponding t, $S_u^1$ is the set of knowledge triples that are 1-hop(s) away from seed set $V_u$, $o_u^1$ means the result of the user interest after the 1-hop:

$$o_u^1 = \sum_{(h_i r_i, t_i) \in S_u^1} p_i t_i \tag{11}$$

The procedure can be performed iteratively on user u's ripple sets $S_u^i$ for $i = 1, \ldots, H$ and the result of user embedding $\mathbf{u}$ is:

$$\mathbf{u} = o_u^1 + o_u^2 + \cdots + o_u^H \tag{12}$$

Given user u's raw feature vector $\mathbf{u}$, we use an L-layer MLP to extract the feature of user u, $M^K$ is a fully connected neural network with K layer:

$$U_L = M^L(\mathbf{u}) \tag{13}$$

For item v, we use L cross and compress units to extract its latent feature:

$$V_L = E_{e \sim S(v)}[C^L(v, e)[v]] \tag{14}$$

where S(v) is the set of associated entities of item v. After acquiring the latent feature of user u and item v, the final predicted probability of user u engaging with item v can be obtained through the prediction function:

$$\hat{y}_{uv} = \sigma(U_L^T V_L) \tag{15}$$

We can obtain the final representation by formulas (7)~(12):

$$\hat{y}_{uv} = \sigma \left( M^L \left( \sum_{n=1}^{H} \sum_{(h_i r_i, t_i) \in S_u^n} \frac{\exp(v^T R_i h_i) t_i}{\sum_{(h,r,t) \in S_u^1} \exp(v^T R_i h_i)} \right)^T, V_L \right) \tag{16}$$

where $\sigma(x) = \frac{1}{1+\exp(-x)}$ is the sigmoid function.

## 5 Learning algorithm

The complete loss function of Ripp-MKR is as follows:

$$\mathcal{L} = \mathcal{L}_{RS} + \mathcal{L}_{KG} + \mathcal{L}_{REG} \tag{17}$$

The loss of Ripp-MKR consists of three parts, namely, the loss of the recommendation module, the loss function of the KGE module and the regularization term for preventing overfitting.

$$\mathcal{L}_{RS} = \sum_{u \in U, v \in V} \mathcal{F}(\hat{y}_{uv}, y_{uv})$$

$$\mathcal{L}_{RS} = \sum_{u \in U, v \in V} -(y_{uv} \log \sigma(u^T v) + (1 - y_{uv}) \log(1 - \sigma(u^T v))) \tag{18}$$

For the loss in the recommendation module, u and v represent the sets of users, and items $\mathcal{F}$ are the cross-entropy function. σ is the sigmoid function.

$$\mathcal{L}_{KG} = \lambda_1 \left( \sum_{(h',r,t') \notin G} score(h', r, t') - \sum_{(h,r,t) \in G} score(h, r, t) \right) \tag{19}$$

For the loss in the KGE module, the main method of convergence is to increase the score for all true triples, while reducing the score for all false triples. $\lambda_1$ is the balancing parameter for the KGE module.

$$\mathcal{L}_{REG} = \lambda_2 \|w\|_2^2 + \frac{\lambda_3}{2} \left( \|V\|_2^2 + \|E\|_2^2 + \sum_{r \in R} \|R\|_2^2 \right) \tag{20}$$

$\mathcal{L}_{REG}$ is the regularization term. It consists mainly of two parts, $\lambda_2$ and $\lambda_3$, which are the balancing parameters. Now the learning algorithm for Ripp-MKR is introduced. The Learning algorithm for Ripp-MKR is shown as Table 2.

## 6 Experiments

This section will evaluate the Ripp-MKR model based on three realistic scenarios: movies, books, and music recommendations. This chapter introduces the data set, baseline, and experimental parameter settings. Then, the experimental results and the analysis of the results are presented.

### 6.1 Dataset

The data set used in this article is shown below.

a. **MovieLens-1M** [39] is a commonly used recommender system dataset that mainly includes user data, movie data, and rating data, which consists of approximately 1 million explicit ratings (ranging from 1 to 5) on the MovieLens website. The knowledge graph section dataset contains the movie's attributes and tags.

b. **Book-Crossing** [40] dataset contains 1,149,780 explicit ratings (ranging from 0 to 10) of books in the Book-Crossing community. The dataset contains binary feedback between

**Table 2. Learning algorithm for Ripp-MKR.**

| **Algorithm 1** Learning algorithm for Ripp-MKR |
|---|
| Input: Interaction matrix **Y**, knowledge graph G |
| Output: Prediction function F ($u$,$v$|Θ, **Y**, G) |
| 1: Initialize all parameters |
| 2: Calculate the ripple sets for each user $\boldsymbol{u}$ on Eqs (7)–(9) |
| 3: For number of training iterations do: /Recommendation task |
| 4: For $t$ steps do: |
| 5: Sample minibatch of positive and negative interactions from Y; |
| 6: Sample $e \sim S(v)$ for each item $v$ in the minibatch |
| 7: Calculate the gradients on the minibatch by back propagation according to Eqs (1)–(4) and Eqs (10)–(16) |
| 8: End for |
| 9: Sample the minibatch of true and false triples from $G$ |
| 10: Sample $v \sim S(h)$ for each head $h$ in the minibatch; |
| 11: Update the parameters of F by gradient descent on Eqs (1)–(4), (17)–(20); |
| 12: End for |

users and books, and the KG for each dataset is built by mapping books to the corresponding entities in Satori, DBpedia, or Freebase.

c. **Last.FM** [41] is the most popular dataset for music recommendation. The dataset comprises information about users and their music listening records from the Last.FM online music system.

Because MovieLens-1M and Book-Crossing contain explicit feedback data, we convert their data to implicit feedback data; each item that is marked as 1 indicates that the user has rated the item (the threshold of rating is 4 for MovieLens-1M, while no threshold is set for Book-Crossing due to its sparsity), and each sample looked at by each user has a UN tag set to 0, this is a rating of the same size. An unwatched set is sampled for each user; each sample is marked as 0 and is of the same size as the watched set.

A similar approach is used for the experimental process of the RippleNet model. We use Microsoft Satori to construct the knowledge graph for each dataset. For MovieLens-1M and Book-Crossing, we first select a subset of triples from the whole KG whose relation name contains "movie" or "book" and whose confidence level is more significant than 0.9. Given the sub-KG, we collect the IDs of all valid movies/books by matching their names with the tail of the triples (head, film.film.name, tail) or (head, book.book.title, tail). For simplicity, items with no matched or multiple matched entities are excluded. We then match the IDs with the head and tail of all KG triples, select all well-matched triples from the sub-KG, and extend the set of entities iteratively up to four hops. The basic statistics settings for the three datasets is shown as Table 3.

## 6.2 Baseline

To demonstrate the reliability of our algorithms, we use other models that incorporated knowledge graph techniques as a baseline. These models and the Ripp-MKR model proposed in this paper jointly use the same dataset to conduct experimental verification in the sense of the AUC value and ACC value of the model.

**MKR** [35]: This model is the basis of the Ripp-MKR model proposed in this paper. We set the number of high-level layers to K = 1, $f_{RS}$ is the inner product, and $\lambda_2 = 10^{-6}$, L =1, d = 8 and $\lambda_1 = 0.5$. In this model, all attributes are added to the KGE unit, training is conducted in the knowledge graph unit, and only project-user rating is used as the training input in the RS unit. $\lambda_1$ is the weight of $L_1$ regularization and $\lambda_2$ is the weight of $L_2$ regularization. L is the number of low layers.

**PER** [36]: PER treats the KG as a heterogeneous information network and extracts meta-path-based features to represent the connectivity between users and items. In this paper, we use manually designed user-item-attribute-item paths as features.

**DKN** [26]: DKN makes use of entity embedding and word embedding as multiple channels and combines them in a CNN for CTR prediction. In this paper, we use movie names as the textual input for DKN. The dimension of word embedding and entity embedding is 64, and the number of filters is 128 for window sizes 1, 2, and 3. Although DKN is an in-depth

**Table 3. Basic statistics settings for the three datasets.**

| Dataset | users | items | interactions | KG triples |
|---|---|---|---|---|
| MovieLens-1M | 6,036 | 2,347 | 753772 | 20195 |
| Book-Crossing | 17860 | 14910 | 139746 | 19793 |
| Last.FM | 1872 | 3846 | 42346 | 15518 |

recommendation model designed for news, it represents the combination of knowledge graph and recommendation system.

**Wide&Deep** [21]: This is a deep recommendation model that combines a (wide) linear channel with a (deep) nonlinear channel. We concatenate the raw features of users and items as well as the corresponding averaged entity embeddings learned from TransR as input. The dimensions of the user, item, and entity are 64, and we use a two-layer deep channel with dimensions of 100 and 50 as well as a wide channel.

**RippleNet** [7]: This model is the basis of the Ripp-MKR model proposed in this paper. The parameters we set for the dataset MovieLens-1M are as follows: $d = 16$, $H = 2$, $\lambda_1 = 10^{-7}$, $\lambda_2 = 0.01$, and $\eta = 0.02$. The parameters we set for the dataset are as follows: $d = 4$, $H = 3$, $\lambda_1 = 10^{-5}$, $\lambda_2 = 0.01$, and $\eta = 0.001$. The hyperparameter settings for Last.FM are $d = 8$, $H = 2$, $\lambda_1 = 10^{-6}$, $\lambda_2 = 0.01$, and $\eta = 0.02$. $\lambda_1$ is the weight of $L_1$ regularization and $\lambda_2$ is the weight of $L_2$ regularization. H is the maximum hops.

## 6.3 Experiments setup

In Ripp-MKR, we set the number of high-level layers to $K = 1$, $f_{RS}$ is the inner product, and $\lambda_2 = 10^{-6}$ for all three datasets, and other hyperparameters are given in Table 4. The settings of the hyperparameters are determined by optimizing the AUC on a validation set. For each dataset, the ratio of the training, validation, and test sets is 6:2:2. Each experiment is repeated three times, and the average performance is reported. We evaluate our method in two experimental scenarios: (1) In click-through rate (CTR) prediction, we apply the trained model to each piece of interaction in the test set and output the predicted click probability. We use the AUC and accuracy to evaluate the performance of CTR prediction. (2) In the top-K recommendation, we use the trained model to select K items with the highest predicted click probability for each user in the test set and choose Precision@K and Recall@K to evaluate the recommended sets.

## 6.4 Result

Figs 4 and 5 and Table 5 show the experimental comparison results of the Ripp-MKR model and other baseline models on three different datasets. Evaluation indexes such as the AUC, ACC, Precision@K, and Recall@K were demonstrated. Table 5 shows the results of the AUC and accuracy in CTR prediction.

As shown in Table 5, PER performs poorly on movie, book, and music recommendations because user-defined metapaths can hardly be optimal in reality. Moreover, because the text length in the data set is relatively short, the DKN model results are also unsatisfactory on these three data sets. This is because the DKN model is a recommender model involved in news recommendations. It mainly processes news titles with knowledge graph and conducts sequential training, so it has strong pertinence and limitations, leading to unsatisfactory recommendation results The performance of the Wide&Deep model is not as good as those of the MKR model and the RippleNet model because this model only splices attributes and does not integrate semantic analysis into training as side information like the other two models. For the RippleNet and MKR models, the results are excellent. However, compared with the Ripp-MKR model, there are still some deficiencies. The main reason is that the Ripp-MKR model is the

**Table 4. Basic statistics settings for the three datasets.**

| | |
|---|---|
| MovieLens-1M | $d = 16$, $L = 1$, $H = 2$, $batch\_size = 1024$, $\lambda_1 = 10^{-6}$, $\lambda_2 = 0.01$, $\lambda_3 = 10^{-8}$, $hop = 2$ |
| Book-Crossing | $d = 8$, $L = 1$, $H = 1$, $batch\_size = 16$, $\lambda_1 = 10^{-6}$, $\lambda_2 = 10^{-5}$, $\lambda_3 = 10^{-7}$, $hop = 2$ |
| Last.FM | $d = 4$, $L = 2$, $H = 2$, $batch\_size = 256$, $\lambda_1 = 10^{-6}$, $\lambda_2 = 0.01$, $\lambda_3 = 10^{-7}$, $hop = 2$ |

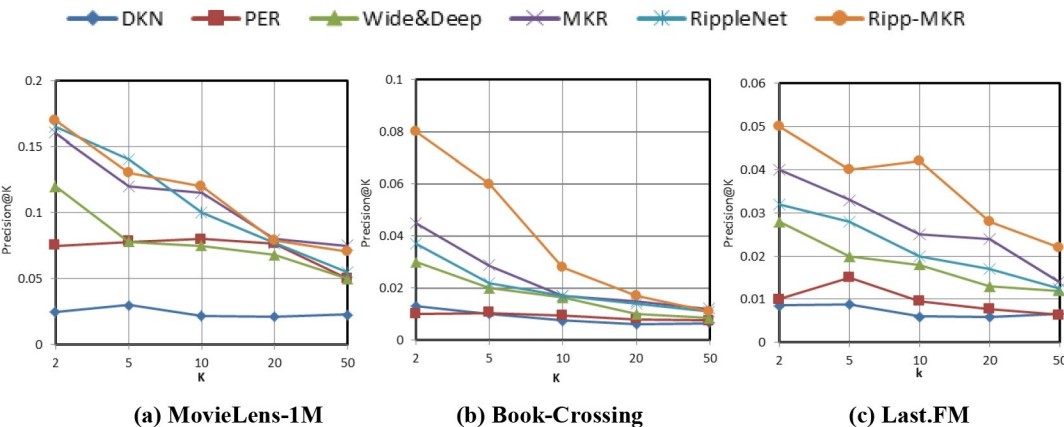

**Fig 4. The results of Precision@K in the top-K recommendation.**

fusion of the RippleNet and MKR models, which makes up for the deficiencies in the architecture of these two models. We can reduce the RippleNet model's sensitivity to the degree of data sparseness by incorporating the idea of compress units. With the concept of ripple propagation, the MKR model can dig deeper into the connections between items and improve the logic of the knowledge map structure. In general, our Ripp-MKR performs best among all methods on the three datasets.

Ripple-MKR also achieves outstanding performance in the top-K recommendation, as shown in Figs 4 and 5.

The size of the ripple set in each hop is discussed next. We vary the size of a user's ripple set in each hop to further investigate RippleNet and Ripp-MKR's robustness. The AUC results for the two datasets are presented in Table 6.

The Ripp-MKR model has the same sensitivity to the ripple set's size as the RippleNet model, and the accuracy increases with increasing set size. With the increase in the ripple set's size, the performance of Ripp-MKR is improved at first because a more extensive ripple set can encode more knowledge from the KG.

We vary the ratio of the training set of MovieLens-1M from 100% to 20% (while the validation and test set remain fixed) and report the results of the AUC in CTR prediction for all

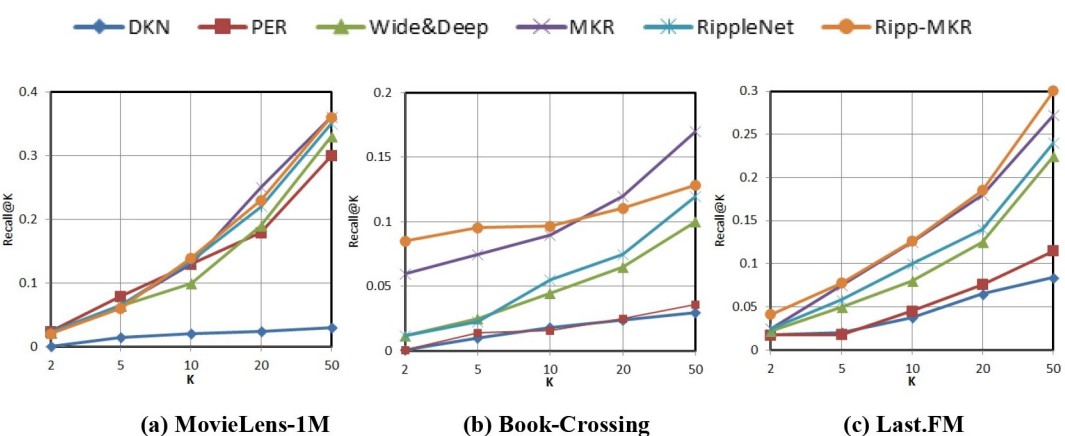

**Fig 5. The results of Recall@K in the top-K recommendation.**

**Table 5. The results of the AUC and accuracy in CTR prediction.**

| MODEL | MovieLens-1M | | Book-Crossing | | Lat.FM | |
|---|---|---|---|---|---|---|
| | *AUC* | *ACC* | *AUC* | *ACC* | *AUC* | *ACC* |
| PER | 0.710 | 0.664 | 0.623 | 0.588 | 0.633 | 0.596 |
| DKN | 0.655 | 0.589 | 0.622 | 0.598 | 0.602 | 0.581 |
| Wide&Deep | 0.898 | 0.820 | 0.712 | 0.624 | 0.756 | 0.688 |
| MKR | 0.917 | 0.843 | 0.734 | 0.704 | 0.797 | 0.752 |
| RippleNet | 0.920 | 0.842 | 0.729 | 0.062 | 0.768 | 0.691 |
| Ripp-MKR | 0.922 | 0.845 | 0.740 | 0.712 | 0.799 | 0.756 |

**Table 6. The results of the AUC w.r.t. different sizes of a user's ripple set.**

| MODEL | 2 | | 4 | | 8 | | 16 | | 32 | |
|---|---|---|---|---|---|---|---|---|---|---|
| | Ripp-MKR | RippleNet | Ripp-MKR | RippleNet | Ripp-MKR | RippleNet | Ripp-MKR | RippleNet | Ripp-MKR | RippleNet |
| MovieLens-1M | 0.9030 | 0.9030 | 0.910 | 0.908 | 0.916 | 0.911 | 0.917 | 0.918 | **0.922** | 0.920 |
| Book-Crossing | 0.7236 | 0.659 | 0.728 | 0.696 | 0.730 | 0.708 | 0.733 | 0.726 | **0.740** | 0.729 |

**Table 7. Results of the AUC on MovieLens-1M in CTR prediction with different ratios of training set r.**

| MODEL | 20% | 40% | 60% | 80% | 100% |
|---|---|---|---|---|---|
| PER | 0.607 | 0.638 | 0.663 | 0.688 | 0.710 |
| DKN | 0.582 | 0.601 | 0.620 | 0.638 | 0.655 |
| Wide&Deep | 0.802 | 0.815 | 0.840 | 0.876 | 0.898 |
| MKR | 0.874 | 0.882 | 0.897 | 0.908 | 0.917 |
| RippleNet | 0.851 | 0.862 | 0.878 | 0.901 | 0.920 |
| Ripp-MKR | 0.881 | 0.887 | 0.898 | 0.910 | 0.922 |

methods. The results are shown in Table 7. We observe that the performance of all methods deteriorates with a reduction in the training set. The results show that in less training data, the algorithm proposed in this paper is still superior to other models.

## 7 Conclusion and future work

This paper proposes Ripp-MKR, an end-to-end framework that naturally incorporates knowledge graphs into recommender systems by combining joint training with alternating training. Combined with the MKR model and the RippleNet model's main ideas, the cross and compress unit of MKR is retained, and the ripples propagation idea is used to represent users' features.

Ripp-MKR overcomes the limitations of the existing embedding-based and path-based KG-aware recommendation methods by introducing preference propagation, which automatically propagates users' potential preferences and explores their hierarchical interests in the KG. The embedding operation of the MKR model for users is disabled and realized through the process of ripple propagation.

We conduct extensive experiments in three recommendation scenarios. The results demonstrate the significant superiority of Ripp-MKR over strong baselines. For future work, we plan to (1) integrate knowledge graph training methods and (2) design a model to better explore users' potential interests and improve the performance of the knowledge graph.

## Acknowledgments

We wish to thank Hao Wang (Jilin University) to prepare the figures, NingWang for insightful comments on the manuscript. Besides, our heartfelt thanks also go to Prof. Hao Zhang to make this thesis and his enlightening lectures from which I have benefited a great deal. We are pleased to acknowledge XinTaoMa (JiLin University) for their invaluable assistance throughout the original manuscript's preparation. They graciously make considerable comments and sound suggestions to the outline of this paper. We would like to express our gratitude to all those who helped usus during this thesis writing.

## Author Contributions

**Conceptualization:** YueQun Wang, LiYan Dong.

**Data curation:** YueQun Wang, LiYan Dong.

**Formal analysis:** YueQun Wang, LiYan Dong.

**Investigation:** YueQun Wang.

**Methodology:** YueQun Wang.

**Project administration:** YueQun Wang.

**Resources:** YueQun Wang, LiYan Dong, YongLi Li.

**Software:** YueQun Wang, LiYan Dong, YongLi Li.

**Supervision:** Hao Zhang.

**Validation:** YueQun Wang, YongLi Li.

**Visualization:** YueQun Wang, YongLi Li.

**Writing – original draft:** YueQun Wang.

**Writing – review & editing:** YueQun Wang, Hao Zhang.

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
