## [Decision Letter · Decision Letter 0]

29 Mar 2021

PONE-D-21-07884

Multitask Feature Learning Approach for Knowledge Graph Enhanced Recommendations with RippleNet

PLOS ONE

Dear Dr. Zhang,

Thank you for submitting your manuscript to PLOS ONE. After careful consideration, we feel that it has merit but does not fully meet PLOS ONE’s publication criteria as it currently stands. Therefore, we invite you to submit a revised version of the manuscript that addresses the points raised during the review process.

We look forward to receiving your revised manuscript.

Kind regards,

Qi Zhao

Academic Editor

PLOS ONE

Journal Requirements:

"The funders had no role in study design, data collection and analysis, decision to

publish, or preparation of the manuscript."

4. Please ensure that you refer to Figure 1 in your text as, if accepted, production will need this reference to link the reader to the figure.

Reviewers' comments:

Reviewer's Responses to Questions

**Comments to the Author**

1. Is the manuscript technically sound, and do the data support the conclusions?

Reviewer #1: Yes

Reviewer #2: Yes

Reviewer #3: Yes

2. Has the statistical analysis been performed appropriately and rigorously? 

Reviewer #1: Yes

Reviewer #2: Yes

Reviewer #3: Yes

3. Have the authors made all data underlying the findings in their manuscript fully available?

Reviewer #1: Yes

Reviewer #2: Yes

Reviewer #3: Yes

4. Is the manuscript presented in an intelligible fashion and written in standard English?

Reviewer #1: Yes

Reviewer #2: Yes

Reviewer #3: Yes

5. Review Comments to the Author

Reviewer #1: This paper combines the relevant technologies of depth recommendation and knowledge graph, the knowledge graph as side Information, and combining it with the recommender system in two forms. In this paper, use two models are combined to make up for their shortcomings and maximize the advantages of the models.

(1) In Sec.2, the Related Work are introduced. While displaying the relevant technologies, it is also necessary to clarify why these technologies are used. For example, explain the role of Side information and so on.

(2) As for the illustration of the proposed model, fig2 is not a good one. Firstly, it’s hard to recognize the input of the model and the relation between different modules. Secondly, the meanings of symbols and different color parts are not given.

(3) The same symbol expresses too much meaning, for example, the meaning of L and K in Eq.4 is not elaborated.

(4) The figure presented in this paper needs to be explained as comprehensively as possible. Include the meaning of each module and the links between the modules, should be given as comprehensive as possible

(5) As we know, DKN is designed specially for news recommendation where the entities are extracted from news titles. Obviously less entities could be extracted from movie names since they are often shorter than news titles. These need to be explained in the results. The experiment shows that the accuracy of recommendation is greatly improved。

Reviewer #2: In this paper, the authors have described the Ripp-MKR deep learning recommendation model in detail, the key algorithms of the Ripp-MKR model which combine joint learning and alternate learning methods in KG-aware recommendations. The result indicates that the Ripp -MKR model has significant advantages over other baseline models. The paper is well organized. The introduction gave a satisfactory literature survey on the similar topic and it outlined the proposed method well. On my point of view, the idea of this work is original and interesting to deep network recommendation systems.

(1) There are many notations in this paper, which should be clearly defined:

What do L and K represent respectively in Eq.4?

What do o_u^1 represent respectively in Eq.11?

(2) Some misused words need to be corrected such as:

Recommendation Modul ->Recommendation Module

KGE Modul ->KGE Module

“The relationship structure between the knowledge graph and the recommender system is shown in Tab.1” Should change Tab.1 to Fig.1.

(3) Since Figure 2 is the main structure of the model proposed in this paper, it needs to be introduced emphatically. Figure 2 should clarify the inputs and outputs of the model, the meaning of the symbols and the parts with different colors need to be explained.

(4) The format of the Figure mentioned in the body should be uniform. Cite figures with the format: Fig 1A, Fig 1B, Fig 2, Fig 3, etc. The cite to Figure 2 and Figure 3 in this article is incorrectly formatted. The name of Figure 1 is too long, so there is no need to explain the figure in the title.

(5) In Sec.6, the parameter of baseline should give reasonable meanings. Such as explain the meaning of λ in each baseline model.

Reviewer #3: This paper studies the recommendation system, which aims to incorporate knowledge graph embedding learning into recommendation systems. In order to combine joint learning and alternate learning of knowledge graph into recommendation systems, the authors proposed a multitask feature learning approach for knowledge graph enhanced recommendations with RippleNet. In addition, cross and compress units are used to share latent features and learn the high-order interactions among items in recommendation systems and entities in the knowledge graph. Finally, the model combines the knowledge graph with the historical interaction record of a user’s historical records to represent the users’ characteristics.

Shortcomings of this paper:

(1) There is no need to introduce the details of recommender system (RS) categories, side information, etc. in Sec.1, which should be removed into Sec.2. For the Sec.1, only a few technologies need to be mentioned, mainly explaining the significance and purpose of the model design.

(2)The authors should elaborate more about their solution's motivations and rationales. For example, why did they combine joint learning and alternating learning? What is the advantage of such process compared with previous KG-based RS? What is the objective of corss&compress units?

(3) Notations and explanations should not only be shown in the table, but also explained in terms of the notation after the formula is listed. Although it has been mentioned above, the explanation of important formula symbols still needs to be mentioned again

(4) The writing of this paper is somewhat lacking due to some grammar and format errors and typos So I suggest the authors proof read their manuscript.

(5) Figure 2~5 cannot be found in the paper, only figure title here.

6. PLOS authors have the option to publish the peer review history of their article (what does this mean?). If published, this will include your full peer review and any attached files.

Reviewer #1: **Yes: **Zhen Liu

Reviewer #2: No

Reviewer #3: No

---

## [Author Response · Author response to Decision Letter 0]

11 Apr 2021

Reviewer#1, Concern # 1: In Sec.2, the Related Work are introduced. While displaying the relevant technologies, it is also necessary to clarify why these technologies are used. For example, explain the role of Side information and so on.

Author response: Thank the reviewers for your comments. As for the design idea and concept of the model proposed in this paper, there is not much elaboration in this paper's second chapter. We will give a reasonable explanation in the first section of the second chapter of this paper.

Author action: We updated the manuscript by giving the reasons and technical basis for combining the model and knowledge graph as side information in Sec.2.

Reviewer#1, Concern # 2: As for the illustration of the proposed model, fig2 is not a good one. Firstly, it’s hard to recognize the input of the model and the relation between different modules. Secondly, the meanings of symbols and different color parts are not given.

Author response: Thank the reviewers for your comments. We are sorry that the model presented in Figure 2 is not explained in-depth in this paper.

Author action: We updated the manuscript by explaining Fig. 2, including the meaning of each color module and the fusion process of the knowledge graph and recommendation system.________________________________________

Reviewer#1, Concern # 3: The same symbol expresses too much meaning, for example, the meaning of L and K in Eq.4 is not elaborated.

Author response: Thank you for your comments. This issue has been modified and the full text reviewed.

Author action: We updated the manuscript by revising the full text for this type of mistake.

Reviewer#1, Concern # 4: The figure presented in this paper needs to be explained as comprehensively as possible. Include the meaning of each module and the links between the modules, should be given as comprehensive as possible

Author response: This is a question which indeed deserves our attention. The interpretation of Figure 2 has been answered and modified in Concern 2, and the full text has been reviewed to give a reasonable explanation of each figure.

Author action: We updated the manuscript by adding an explanation of Figures 2 and 3.

Reviewer#1, Concern # 5: As we know, DKN is designed specially for news recommendation where the entities are extracted from news titles. Obviously less entities could be extracted from movie names since they are often shorter than news titles. These need to be explained in the results. The experiment shows that the accuracy of recommendation is greatly improved.

Author response: Thank you for your comments. DKN is a depth model designed for news recommendations. This paper introduces DKN into baseline because DKN is also a model that combines depth recommendation with a knowledge graph. Besides, the MKR model and RippleNet model are introduced as baseline. Although DKN is a depth model with limited application, it is also representative. We will explain and indicate the limitations of DKN in the experimental results section and the reasons for using DKN as baseline

Author action: We updated the manuscript by explaining the limitations of DKN and the reasons for using DKN as the baseline.________________________________________

Reviewer#2, Concern # 1: There are many notations in this paper, which should be clearly defined: What do L and K represent respectively in Eq.4?What do o_u^1 represent respectively in Eq.11?

Author response: Thank you for your comments. We have revised the above issues and reviewed the full text to revise the same issues

Author action: We updated the manuscript by giving an explanation of all the symbols.

Reviewer#2, Concern # 2: Some misused words need to be corrected such as:Recommendation Modul ->Recommendation Module;KGE Modul ->KGE Module;“The relationship structure between the knowledge graph and the recommender system is shown in Tab.1” Should change Tab.1 to Fig.1.

Author response: Thank you for your comments. We have revised the above issues and reviewed the full text to revise the same issues.

Author action: We updated the manuscript by modifying and checking the errors mentioned above.

Reviewer#2, Concern # 3: Since Figure 2 is the main structure of the model proposed in this paper, it needs to be introduced emphatically. Figure 2 should clarify the inputs and outputs of the model, the meaning of the symbols and the parts with different colors need to be explained.

Author response: Thank the reviewers for your comments. We are sorry that the model presented in Figure 2 is not explained in-depth in this paper.

Author action: We updated the manuscript by explaining Fig. 2, including the meaning of each color module and the fusion process of the knowledge graph and recommendation system.________________________________________

Reviewer#2, Concern # 4: The format of the Figure mentioned in the body should be uniform. Cite figures with the format: Fig 1A, Fig 1B, Fig 2, Fig 3, etc. The cite to Figure 2 and Figure 3 in this article is incorrectly formatted. The name of Figure 1 is too long, so there is no need to explain the figure in the title.

Author response: We are grateful to the reviewers for their careful review of my paper. The referencing of the figures has been consolidated and the title of each figures has been modified.

Author action: We updated the reference format by unifying the form and changed the title of Figure 1

Reviewer#2, Concern # 5: In Sec.6, the parameter of baseline should give reasonable meanings. Such as explain the meaning of λ in each baseline model.

Author response: Thank you for your comments. We have revised the above issues and reviewed the full text to revise the same issues.

Author action: We updated the manuscript by giving the parameter of baseline should give reasonable meanings in Sec.6.________________________________________

Reviewer#3, Concern # 1: There is no need to introduce the details of recommender system (RS) categories, side information, etc. in Sec.1, which should be removed into Sec.2. For the Sec.1, only a few technologies need to be mentioned, mainly explaining the significance and purpose of the model design.

Author response: We are grateful to the reviewers for their careful review of my paper. We will revise it according to your suggestion.

Author action: We updated the manuscript by modifying the contents of Chapter 1 and Chapter 2 so that Chapter 2 highlights the technical issues and Chapter 1 shows the basic ideas of the model

Reviewer#3, Concern # 2: The authors should elaborate more about their solution's motivations and rationales. For example, why did they combine joint learning and alternating learning? What is the advantage of such process compared with previous KG-based RS? What is the objective of corss&compress units?

Author response: Thank the reviewers for their opinions. We will add a reasonable explanation to the model idea in the first chapter

Author action: We updated the manuscript by elaborating more about our solution's motivations and rationales in Sec .1.

Reviewer#3, Concern # 3: Notations and explanations should not only be shown in the table, but also explained in terms of the notation after the formula is listed. Although it has been mentioned above, the explanation of important formula symbols still needs to be mentioned again.

Author response: We are grateful to the reviewers for their careful review of my paper. This issue has been modified and the full text reviewed.

Author action: We updated the manuscript by revising the full text for this type of mistake.

Reviewer#3, Concern # 4: The writing of this paper is somewhat lacking due to some grammar and format errors and typos So I suggest the authors proof read their manuscript.

Author response: Thank you for your careful review of our article. For grammar problems, we will review the full text and correct grammatical errors.

Author action: We updated the manuscript by correcting any grammatical errors in the text.

Reviewer#3, Concern # 5: Figure 2~5 cannot be found in the paper, only figure title here.

Author response: Thank the reviewers for their opinions. During the submission process, the figure is submitted separately from the paper, the figure at the end of the Manuscript.

Author action: We updated the manuscript by doing nothing.

---

## [Decision Letter · Decision Letter 1]

21 Apr 2021

Multitask Feature Learning Approach for Knowledge Graph Enhanced Recommendations with RippleNet

PONE-D-21-07884R1

Dear Dr. Zhang,

We’re pleased to inform you that your manuscript has been judged scientifically suitable for publication and will be formally accepted for publication once it meets all outstanding technical requirements.

Kind regards,

Qi Zhao

Academic Editor

PLOS ONE

Additional Editor Comments (optional):

Reviewers' comments:

Reviewer's Responses to Questions

**Comments to the Author**

1. If the authors have adequately addressed your comments raised in a previous round of review and you feel that this manuscript is now acceptable for publication, you may indicate that here to bypass the “Comments to the Author” section, enter your conflict of interest statement in the “Confidential to Editor” section, and submit your "Accept" recommendation.

Reviewer #2: All comments have been addressed

Reviewer #3: All comments have been addressed

2. Is the manuscript technically sound, and do the data support the conclusions?

Reviewer #2: Yes

Reviewer #3: Yes

3. Has the statistical analysis been performed appropriately and rigorously? 

Reviewer #2: Yes

Reviewer #3: Yes

4. Have the authors made all data underlying the findings in their manuscript fully available?

Reviewer #2: Yes

Reviewer #3: Yes

5. Is the manuscript presented in an intelligible fashion and written in standard English?

Reviewer #2: Yes

Reviewer #3: Yes

6. Review Comments to the Author

Reviewer #2: (No Response)

Reviewer #3: I believe the authors have adequately addressed the comments raised in the previous round of review. At this stage, I have no further points to make.

7. PLOS authors have the option to publish the peer review history of their article (what does this mean?). If published, this will include your full peer review and any attached files.

Reviewer #2: No

Reviewer #3: No

---

## [Editor Report · Acceptance letter]

26 Apr 2021

PONE-D-21-07884R1 

Multitask Feature Learning Approach for Knowledge Graph Enhanced Recommendations with RippleNet 

Dear Dr. ZHANG:

I'm pleased to inform you that your manuscript has been deemed suitable for publication in PLOS ONE. Congratulations! Your manuscript is now with our production department. 

Kind regards, 

on behalf of

Dr. Qi Zhao 

Academic Editor

PLOS ONE